# Mask Models are Token Level Contrastive Learners

## Abstract

In recent years, the field of self-supervised learning has seen a surge in the development of mask models, which have been demonstrated to have strong performance on downstream tasks and efficient training. To better understand the underlying mechanism behind these models' success, we propose a theoretical framework for understanding mask models. By treating mask modeling as a low-rank recovery task, we demonstrate that it is a parametric version of Spectral Clustering and the reconstruction loss conforms to the form of Spectral Contrastive loss. This means that mask modeling can be understood as a token level Contrastive Learning. Our framework can be used to explain why optimal masking ratios vary among modalities and why there is a large gap between linear probing and finetuning performance for mask models. Additionally, our analysis suggests that the success of mask models depends on the model architecture, where a token mixing layer and layer normalization are crucial for the success of mask models. Our framework has the potential to be a step stone for future algorithm and network architecture design in the field of self-supervised learning.

## 1 Introduction

With the rapid-growth of deep learning and its increasing demand for data, self-supervised learning arises as a research topic in-demand. Among the successful self-supervised learning models, mask models have received significant attention for their strong downstream performance and efficient training [15, 38, 12, 5, 23, 34, 18, 4]. However, mask-modeling has long been regarded as an engineering trick, and its underlying mechanism remains poorly understood.

Empirically, we have observed that Mask Image Modelling (MIM) leads to varying performance improvements on different downstream tasks compared to previous baselines [15]. Specifically, MIM has been found to perform better on fine-grained tasks such as semantic segmentation and object detection, compared to classification tasks. This phenomenon leads us to hypothesize that the representation learned by MIM models is fundamentally similar to that of image segmentation (clustering).

In this work, we propose a theoretical analysis of mask modelling by treating it as a low-rank recovery (LRR) task. Our analysis further demonstrates that the reconstruction loss can be rewritten as a Contrastive loss [10].

The LRR problem aims to find the low-rank approximation of a given matrix, and has been used as a method for subspace clustering [28]. Additionally, as the optimal solution of the LRR problem is a combination of leading eigenvectors, we are naturally led to Spectral Clustering, which also utilizes leading eigenvectors [32, 26]. Our results show that MIM approximates the Spectral Clustering features of an image-related graph, where each node represents a patch of the image.

By viewing the Masked Image Model (MIM) as a parametric version of Spectral Clustering, we can rewrite the reconstruction loss of mask models in the form of Spectral Contrastive loss on the token

level [14].. This allows MIM to be viewed as a token-wise Contrastive Learning method, which attracts similar patches while repelling dissimilar ones, resulting in smaller distances within clusters and larger distances between clusters. However, there are some key differences between mask models and traditional Contrastive Learning methods. Specifically, mask models operate on the token level, whereas traditional Contrastive Learning methods focus on the global feature of the entire input, and in mask models, positive samples are not clearly defined, but are "randomly sampled" based on the similarity between Spectral Clustering features.

Based on the formulation, we could answer several concerning questions about mask models: 1) Why optimal masking ratio vary among modalities? 2) Why is there a large gap between linear probing and finetuning performance for mask models? 3) Does mask modelling rely on network architectures?

For the first question, we argue that a critical factor that affects the goodness of pretrained features is the number of clusters in Spectral Clustering. For example, if we have an image with a dog on the grass, intuitively we should have two clusters: grass and dog. It could be less representative if we have more clusters and divide one of the existing clusters into different sub-clusters and repel one from each other. The number of clusters is given by the number of leading eigenvectors, which is related to the rank of reconstructed matrix in LRR problem and masking ratio in MIMs. This explains why we need different masking ratios in different modalities [12, 15, 34, 18].

For the second question, it is due to the nature of token level Contrastive learning. Pretrained mask models learns to divide tokens into clusters, but doesn't always learn which cluster is most related to the class. Therefore, token mixing layers are needed to "select" clusters. Most MIMs apply an extra BatchNorm layer when performing linear probing, otherwise a huge accuracy drop is witnessed [19, 15]. It could be due to the lack of patch selection and a BatchNorm is needed to add non-linearity. In contrast, we found that partially finetuning one linear layer for row mixing with the prediction head could much improve classification accuracy.

For the third question, the answer is "Yes". Model architectures containing token mixing layers plays a crucial role in the success of mask models in classification tasks [35, 13, 24, 33]. Finetuning these layers allows the model to learn how to select tokens. Meanwhile, the layer normalization in the decoder might also be important, as it serves as a token level batch normalization, which is commonly used in the projection layer of Contrastive Learning models to improve performance [3, 19]. Therefore, we conclude that mask model is dependant of network architecture.

In a summary, our main contributions are:

1. We created a mathematical framework for mask image modeling by viewing it as a low-rank recovery problem.

2. We found that mask model could be viewed as a token level Contrastive Learning, which could account for its good performance on downstream tasks.

3. Our analysis framework could explain several important behaviors of mask models and guide future model architecture design and parameter choosing.

We mainly conducted experiments on images, but our findings could be easily generalized to all modalities.

## 2 Related Works

### 2.1 Mask Image Modelling

The recent trend in self-supervised learning is to train vision transformers using masked images to reconstruct the original ones. [13]. RDifferent types of reconstruction objectives, such as token-wise, feature-wise, and pixel-wise reconstruction, are being tested. [39, 8, 7]. These kinds of pretraining tasks are called Masked Image Modeling (MIM) [5]. There are two main architectures for these models: one that only accesses visible tokens in the encoder and attaches an extra decoder [15, 5], and another that passes both visible and mask tokens into the encoder and has a single linear layer as a decoder [38]. Our formulation is based on the first type of architecture. These mask models serve as a pretrain model, and for downstream tasks, we either finetune or perform linear probing. For classification, a linear head is appended, and the parameters are initialized from the pretrain models. The difference between finetuning and linear probing is that the parameters of the pretrained model are frozen in linear probing.

## 2.2 Theoretical Analysis of Mask Models

Previous works on mask models have provided theoretical frameworks for understanding the attention operation in the encoder [6], proposed that MIMs are learning semantics [27], proved a downstream performance bound for linear probing with MSE loss [22], and claimed that mask models learn global features that are occlusion invariant [20]. Our work is distinct from these previous works in that we emphasize the connection between mask modeling and Contrastive Learning. One work also mentioned that the decoder in MIMs is performing low-rank recovery, but the authors did not link this to the success of MIMs [6].

## 2.3 Spectral Contrastive Loss

The Spectral Contrastive loss was proposed as a way to provide a provable guarantee for downstream task performance [14, 2]. However, some later work has identified issues with the formulation and stronger assumptions are needed to achieve the guarantee [29]. Despite this, the theoretical framework that connects Contrastive Learning and Spectral Clustering is still attractive. Our work is inspired by this analysis framework, but with several differences. In their work, the graph used for Spectral Clustering is inherent, and the authors argue that matrix factorization approximates the node representations of the graph. Our work, instead, explicitly writes out the adjacency matrix of a graph and shows that it is related to the MIM problem. Additionally, our work highlights the importance of rank, which is often overlooked in previous works.

# 3 Preliminary and Notations

## 3.1 Notations of Masked Autoencoder

Our analysis mainly focus on Masked Autoencoder (MAE) style encoder-decoder structure, where the input size of encoder is smaller than that of decoder [15]. Denote the encoder in the mask modeling by $f$, and the decoder by $g$, the sampled visible subset by $X$, and the masked part of the original image by $X_0$. We adopt the Transformer architecture as backbone, where $f$ and $g$ don't change the shape of inputs [35].

**Definition 3.1.** To train the masked autoencoder and achieve the best performance can be interpreted as solving the minimization problem:

$$\underset{f, g, X}{\mathrm{argmin}} \|g \circ f(X) - X_0\|_F^2, \tag{1}$$

where $X \in \mathbb{R}^{N \times F}$, $X \in \mathbb{R}^{N_0 \times F}$. We reshape the matrix of image so that $N$ is the number of visible patches and $N_0$ is the number of masked patches. We also have the loss defined as

$$\mathcal{L}_{MAE}(f, g, X) = \|g \circ f(X) - X_0\|_F^2 \tag{2}$$

**Definition 3.2.** We further define the token mixing layer with weight $W \in \mathbb{R}^{N \times N}$, which mixes features on the patch level. of inputs. In transformers, the layer is the softmaxed query-key matrix.

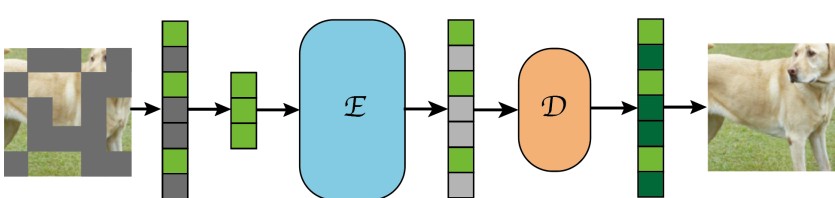

Figure 1: **Overall structure of Mask Autoencoders.** Only visible tokens are passed through the encoder, and a decoder applies encoder features to reconstruct mask tokens.

## 3.2 Basic Low-rank Recovery Problem

The basic low-rank recovery problem solves the following optimization problem:

$$\underset{\widehat{D}}{\mathrm{argmin}} \ \|\widehat{D} - D\|_F \ \text{ subject to } \ \mathrm{rank}(\widehat{D}) \leq \mathrm{rank}(D) \tag{3}$$

Based on the Eckart–Young–Mirsky theorem [17], the low-rank approximation problem has a solution in terms of singular value decomposition of the original matrix, which is in the form: $\widehat{D} = \sum_{i=1}^{r} \sigma_i u_i v_i^\top$, where $\sigma_i$ is the $i^{th}$ singular and $u_i$ and $v_i$ are its corresponding left and right singular vectors. We could also write it as $\widehat{D} = U_r \Sigma_r V_r^\top$, where $U_r$, $V_r$ contains the first $r$ columns of $U$ and $V$, and $\Sigma_r$ is an $r \times r$ matrix with the top $r$ leading singular values as diagonal.

## 3.3 Spectral Clustering with Normalized Adjacency Matrix

Suppose we have a n-node graph $G$ with the adjacency matrix $A$:

$$A = \begin{bmatrix} w_{11} & \cdots & w_{1n} \\ \vdots & \ddots & \vdots \\ w_{n1} & \cdots & w_{nn} \end{bmatrix} \tag{4}$$

The normalized adjacency matrix is defined as $\mathscr{A} = D^{-\frac{1}{2}} A D^{-\frac{1}{2}}$, where $D$ is the degree matrix of graph $G$, which is a diagonal matrix such that $D_{ii} = \sum_{j=1}^{n} w_{ij} = w_i$. To get $k$ cluster of the graph, take the leading top $k$ eigenvectors of $\mathscr{A}$ as node embedding and perform k-means algorithm on the node embedding [26].

# 4 Mask Modeling as a Patch-wise Contrastive Learning

## 4.1 Mask Modeling is Low Rank Recovery

In this section, we formalize MAE as a low-rank recovery task. As $X$ is a smaller portion of the original image and $f$ doesn't change the size of input, $f(X)$ naturally has a lower rank compared to $X_0$. we assume the following condition is true:

**Assumption 4.1.** $g \circ f(X)$ has a lower rank than $X_0$.

*Remark* 4.2. This assumption follows Lemma 6.1 in [6], where they prove an upper bound for the reconstruction with a low rank decoder assumption. Also in practice, even if $g$ is a non-linear function that doesn't guarantee low rank assumption, we find that $g(f(X))$ still has a very low rank. Arguably it is because reconstructing unseen tokens is very hard to optimize and only leading singular vectors can be approximated.

Under this assumption, the minimization problem can be rewritten as

$$\underset{\boldsymbol{f,g,X}}{\operatorname{argmin}} \|g \circ f(X) - X_0\|_F^2 \tag{5}$$
$$\text{subject to} \quad \operatorname{rank}(g \circ f) < \operatorname{rank}(X_0).$$

## 4.2 Mask Modeling is a Parametric Version of Spectral Clustering

In section 3.2, it is showed that the low-rank approximation problem is solved by singular value decomposition of the higher-ranked matrix. Suppose the required rank is $k$, then the optimal solution is a linear combination of top $k$ eigenvectors of $X_0 X_0^\top$ obtained from singular value decomposition.

Consider Spectral Clustering which clusters a graph into $k$ connected components such that there is minimal effect on graph Laplacian. Spectral clustering performs dimensional reduction with $k$ eigenvectors corresponding with the largest $k$ eigenvalues of the normalized adjacency matrix.

Both mask modeling and Spectral Clustering are utilize $k$ eigenvectors, hence, we propose that the behaviors of mask modeling is similar to the behaviors of Spectral Clustering. Consequently, the classifier trained based on mask modeling based $f$ and Spectral Clustering based $f$ gives the same prediction. Formally we have:

**Theorem 4.3.** *Define weights of adjacency matrix for graph $G$ as $w_{ij} = \langle X_{0r,i}, X_{0r,j} \rangle$, where $X_{0r,i}$ is the low-rank approximation of the representation for $i^{th}$ patch of $X_0$. Given the corresponding normalized adjacency matrix $\mathscr{A}$, optimizing mask modeling is equivalent to optimize the following loss on classification downstream tasks.*

$$\mathcal{L}_{spec}(f, g, X) = \left\| (g \circ f(X))(g \circ f(X))^\top - \mathscr{A} \right\|_F^2 \tag{6}$$

*Proof.* The SVD of $X_0$ gives $X_0 = U\Sigma V^\top$, then $A = X_{0r}X_{0r}^\top = U_r\Sigma_r^2 U_r^\top$.

Since $A$ is symmetric and $D$ is diagonal, $\mathscr{A}$ is symmetric and SVD of $\mathscr{A}$ has the form of $U_{\mathscr{A}}\Sigma_{\mathscr{A}}U_{\mathscr{A}}^\top$. Plug in A gives

$$\mathscr{A} = D^{-\frac{1}{2}}AD^{-\frac{1}{2}}$$
$$= D^{-\frac{1}{2}}U_r\Sigma_r^2 U_r^\top D^{-\frac{1}{2}}$$
$$= \left(D^{-\frac{1}{2}}U_r D^{\frac{1}{2}}\right)\left(D^{-\frac{1}{2}}\Sigma_r^2 D^{-\frac{1}{2}}\right)\left(D^{-\frac{1}{2}}U_r D^{\frac{1}{2}}\right)^\top.$$

Therefore, $U_{\mathscr{A}} = D^{-\frac{1}{2}}U_r D^{\frac{1}{2}}$ and $\Sigma_{\mathscr{A}} = D^{-\frac{1}{2}}\Sigma_r^2 D^{-\frac{1}{2}}$.

The SVD of $X_0$ can be rewritten as $X_{0r} = D^{\frac{1}{2}}U_{\mathscr{A}}D^{-\frac{1}{2}}\Sigma_r V_r^\top$. With Eckart–Young–Mirsky Theorem, we rewrite the minimization problem of mask modeling as

$$\underset{f,g,X}{\operatorname{argmin}}\left\| g\circ f(X) - D^{\frac{1}{2}}U_{\mathscr{A}}D^{-\frac{1}{2}}\Sigma_r V_r^\top\right\|_F^2, \tag{7}$$

whose optimal solution is $D^{\frac{1}{2}}U_{\mathscr{A}}D^{-\frac{1}{2}}\Sigma_r V_r^\top$. Note that $D$ is a diagonal matrix, so we could use a decoder to eliminate this term, by setting $g' = D^{-\frac{1}{2}}g$. Therefore, we can discard $D^{\frac{1}{2}}$, making the optimization problem into:

$$\underset{f,g,X}{\operatorname{argmin}}\left\| g\circ f(X) - U_{\mathscr{A}}D^{-\frac{1}{2}}\Sigma_r V_r^\top\right\|_F^2. \tag{8}$$

With some linear algebra calculation, we could further show that the right hand side of Equation 6 is bounded by the error of Equation 8 with big O notation, i.e. given $\left\| g\circ f(X) - U_{\mathscr{A}}D^{-\frac{1}{2}}\Sigma_r V_r^\top\right\|_F^2 \leq \varepsilon$, $\left\|(g\circ f(X))(g\circ f(X))^\top - \mathscr{A}\right\|_F^2 \leq C\varepsilon$ for some constant C.

Let $M = g\circ f(X)$, $N = U_{\mathscr{A}}D^{-\frac{1}{2}}\Sigma_r V_r^\top$, then

$$\left\|(g\circ f(X))(g\circ f(X))^\top - \mathscr{A}\right\|_F^2 = \left\|(g\circ f(X))(g\circ f(X))^\top - (U_{\mathscr{A}}D^{-\frac{1}{2}}\Sigma_r V_r^\top)(U_{\mathscr{A}}D^{-\frac{1}{2}}\Sigma_r V_r^\top)^\top\right\|_F^2$$
$$= \left\|MM^\top - NN^\top\right\|_F^2$$
$$= \frac{1}{4}\left\|(M-N)(M^\top + N^\top) + (M+N)(M^\top - N^\top)\right\|_F^2$$
$$\leq \frac{1}{4}\left(\|M-N\|_F\left\|M^\top + N^\top\right\|_F + \|M+N\|_F\left\|M^\top - N^\top\right\|_F\right)^2$$
$$= \|M-N\|_F^2\|M+N\|_F^2$$

Since $N = U_{\mathscr{A}}D^{-\frac{1}{2}}\Sigma_r V_r^\top$ is defined by the original image and $M$ is an approximation to $N$, we could say that $\|M+N\|_F^2$ is bounded by a constant and thus $\left\|(g\circ f(X))(g\circ f(X))^\top - \mathscr{A}\right\|_F^2 \leq C\varepsilon$, and we finish the proof.

$\square$

Therefore, MAE learns to approximate the Spectral Clustering features. We further discuss the importance of having appropriate $k$ in Section 5.1

### 4.3 $L_{spec}$ is a Spectral Contrastive Loss

Rewrite $L_{spec}$, mask modeling can be viewed as a token level Contrastive Learning. We define the $i^{th}$ row of $g\circ f(X)$ as $\sqrt{w_i}u_i$, the predicted patch representation. We could rewrite $L_{spec}$ into,

$$\mathcal{L}_{\text{spec}} = \left\|(g\circ f(X))(g\circ f(X))^\top - \mathscr{A}\right\|_F^2$$
$$= \left\|(g\circ f(X))(g\circ f(X))^\top - D^{-\frac{1}{2}}AD^{-\frac{1}{2}}\right\|_F^2$$
$$= \sum_{i,j}\left(\frac{w_{ij}}{\sqrt{w_i w_j}} - (\sqrt{w_i}u_i)^\top(\sqrt{w_j}u_j)\right)^2$$
$$= \sum_{i,j}\left(\frac{w_{ij}^2}{w_i w_j} - 2w_{ij}u_i^\top u_j + w_i w_j\cdot\left(u_i^\top u_j\right)^2\right) \tag{9}$$

Apply the kernel trick, changing $w_{ij}$ into $W_{ij}$, such that $W_{ij} = \exp(\frac{w_{ij}}{2\sigma^2})$ [16]. With a choice of $\sigma$, we have $W_{ij}$ defined as (or approximates) the probability of $u_i$ and $u_j$ to be a positive pair. Following the notations of Spectral Contrastive Loss, we make Equation (9) into the form of a Contrastive loss [14].

$$\mathcal{L}_{\text{spec}} = \mathcal{L}_{\text{cont}} + \text{ const}, \tag{10}$$

where $\mathcal{L}_{\text{cont}} = -2 \cdot \mathbb{E}_{u,u^+} \left[ u^\top u^+ \right] + \mathbb{E}_{u,u^-} \left[ \left( u^\top u^- \right)^2 \right]$

*Remark* 4.4. In Haochen et al's work, $u^+$ and $u^-$ is defined as positive/negative samples, that has higher/lower probability to be found in $u$'s augmentation space. In mask models, we similarly define $u^+$ as patches having higher similarity to $u$ and $u^-$ having lower similarity to $u$.

The above shows that MAE loss is equivalent to a Contrastive loss on masked tokens. We further show that it inherently perform Contrastive Learning on visible tokens.

**Assumption 4.5.** Denote one of the original patch of the $i^{th}$ masked token as $X_{0,i}$, the predicted feature $u_i$ is a linear combination of features of visible tokens, such that $u_i = \sum_j a_j u'_j$. Assume this transformation is made by decoder $g$.

**Lemma 4.6.** *Optimal $a_j$ is proportional to patch similarity $u'^\top_j X_{0,i}$.*

*Proof.* Consider MAE loss with regularization, we have the optimization problem to reconstruct one patch:

$$\underset{a_j}{\text{argmin}} \left\| \sum_j a_j u'_j - X_{0,i} \right\|_F^2 + \lambda \sum_j a_j^2 \tag{11}$$

where $j = 1, 2, \cdots N$, denoting the visible patch representations, and $\lambda$ is the regularization strength.

Let $A = [a_1, a_2, \cdots, a_N]^\top$, $U'_{:,k} = [u'_{1,k}, u'_{2,k}, \cdots, u'_{N,j}]^\top$ a rank-1 vector with all patches and $k^{th}$ latent feature, and $X_{:,k} = [X_{1,k}, X_{2,k}, \cdots, X_{N,k}]^\top$ similarly defined. $X_{0,i,k}$ is a scalar corresponds to the $i^{th}$ patch and $k^{th}$ latent feature. With Assumption 4.5, we decompose it into $k$ rank-1 components.

Then Equation 12 becomes:

$$\underset{A}{\text{argmin}} \sum_{k=1}^N \left\| A^\top U'_{:,k} - X_{0,i,k} \right\|_2^2 + \sum_{k=1}^N \frac{\lambda}{N} \|A\|_2^2 \tag{12}$$

Apply Sherman-Morrison formula [1, 31], which states

$$\left( X + mn^\top \right)^{-1} = X^{-1} - \frac{X^{-1}mn^\top X^{-1}}{1 + n^\top X^{-1}m} \tag{13}$$

for any invertible matrix $X$ and rank-1 matrix $m$ and $n$.

Then for each $k$, the optimal solution for A would be:

$$\begin{aligned}
\hat{A_k} &= \left( \frac{\lambda}{N} I + U'_{:,k} U'^\top_{:,k} \right)^{-1} U'_{:,k} X_{0,i,k} \\
&= N \left( \frac{I}{\lambda} - \frac{\frac{I}{\lambda} U'_{:,k} U'^\top_{:,k} \frac{I}{\lambda}}{\frac{1}{N} + U'^\top_{:,k} \frac{I}{\lambda} U'_{:,k}} \right) U'_{:,k} X_{0,i,k} \\
&= \frac{N}{\lambda} U'_{:,k} X_{0,i,k} - \frac{N}{\lambda} \frac{U'_{:,k} \left\| U'_{:,k} \right\|_2^2 X_{0,i,k}}{\frac{N}{\lambda} + \left\| U'_{:,k} \right\|_2^2}
\end{aligned} \tag{14}$$

Since $U'$ is an output of a Layer Norm layer, $\left\| U'_{:,k} \right\|_2^2$ is a constant, denote as $C$. Plug in Equation 14, we get the optimal solution for certain $k$:

$$\hat{A_k} = \frac{N^2}{\lambda N + \lambda C} U'_{:,k} X_{0,i,k} \tag{15}$$

Since Equation 12 solves a sum of least square, the optimal $\hat{A}$ is the mean of $\hat{A}_k$s, i.e. $A = \frac{N}{\lambda N + \lambda C} U'^\top X_{0,i}$, and each optimal $a_j$ is given by

$$\hat{a_j} = \frac{N}{\lambda N + \lambda C} u_j'^\top X_{0,i} \tag{16}$$

$\square$

Meanwhile, as there's only one MLP layer between $u_j'$ and $X_j$, we view $u_j'$ as an approximation to $X_j$, thus

$$\hat{a_j} \approx \frac{N}{\lambda N + \lambda C} X_j X_{0,i} \tag{17}$$

which is proportional to the non-parametric patch similarity defined by the original image. Here we see the representation of masked tokens is mainly composed of similar tokens, performing Contrastive Learning on masked tokens inherently performs Contrastive Learning on visible tokens.

*Remark* 4.7. We could view the layer normalization layer in MAE's decoders as a token level batch normalization, and the entire decoder as a non-linear projection layer in Contrastive Learning methods [3].

*Remark* 4.8. Though reconstructing masked tokens make it more complicated and less explainable, it is required as reconstruction visible tokens could lead to a shortcut solution of identity mapping.

# 5 Patch-wise Contrastive Learning Explains Mask Model Behaviors

Based on the theoretical framework proposed, we could explain several parameter choice and architecture design for mask models.

## 5.1 Mask Ratio for Different Modalities

In Section 4.2, we demonstrate that mask models are a parametric version of Spectral Clustering, and they learn to decrease intra-cluster distances while increasing distances between different clusters through Contrastive Learning. Therefore, an appropriate number of clusters is a crucial factor that affects the quality of the features learned. When we consider each cluster has a pseudo-class label, too few or too many classes can both be indistinct when trying to separate the classes. Therefore, we define the following:

**Definition 5.1.** Let $s$ be the ratio of appropriate cluster numbers to total number of tokens. We have

$$s = \frac{num\_cluster}{num\_tokens} = \frac{k}{N_0}, \tag{18}$$

where $k$ is the number of leading eigenvectors in Spectral Clustering, and $N_0$ is the number of tokens for reconstruction.

In mask models, $k$ is subjected to $rank(g \circ f(X))$, which is determined by the number of visible tokens $N$. If we assume $rank(g \circ f(X))$ is proportional to $N$, we have:

$$s \propto \frac{N}{N_0} \tag{19}$$

As $\frac{N}{N_0} = 1 - mask\_ratio$, $s$ is thus determined by the masking ratio.

Intuitively we know that $s$ is smaller for modalities with lower information density, such as video, vice versa. Therefore, we need a higher masking ratio for lower-density modalities and a smaller one for higher-information-density modalities [18, 36, 15, 34].

## 5.2 Linear Probing Mask Image Models

When tuning MIMs on image classification tasks, there is a significant gap between linear probing and finetuning [15]. A trick that is often used to improve linear probing performance is to append a batch normalization layer before the linear head [9]. Without the BN layer, and with an appropriate batch size, the classification accuracy can drop significantly [37].

245 We argue that this is due to the nature of token-level Contrastive Learning. MIMs only learn to create
246 and separate several clusters, but do not learn which cluster is indicative of the class label. It is often
247 the case that the class token from a pretrained MIM does not learn the correct cluster. Therefore,
248 partially finetuning a token mixing layer can greatly boost accuracy [15]. We also argue that the BN
249 layer adds non-linearity that partly serves as a token mixing layer. Therefore, we may need to rethink
250 whether linear probing is a "fair" method to evaluate MIMs.

### 5.3 Network Architecture Matters

252 As discussed in Section 5.2, MIMs do not know how to select important tokens without finetuning.
253 Therefore, a network architecture with token mixing layers is crucial for the success of mask models
254 on classification tasks. Finetuning these layers allows MIMs to understand what are important
255 tokens. Another important architecture design is the number of attention heads. More attention heads
256 generally improves the expressive ability of mask models by offering more choices of candidate
257 clusters.

## 6 Experiments

259 We have verified several of our assumptions and mathematical formulations with MAE models
260 pretrained on Cifar10 and ImageNet-1K (IN-1K) datasets [21, 11]. The model backbones used
261 for Cifar10 and ImageNet are ViT-Tiny and ViT-Base, respectively [13]. For ViT-Base on IN-1K,
262 we followed the settings of MAE [15]. The parameters of ViT-Tiny on Cifar-10 are given in the
263 supplementary materials. We have used the same parameters for linear probing and finetuning, as we
264 found that changing the optimizer of linear probing to AdamW gives better performance [25].

### 6.1 Low-rank Approximation of Different Mask Ratio

266 To verify Assumption 4.1, and our claim in Section 5.1, we computed the average distance of matrix
267 factorization components between the reconstructed image and the original image on the Cifar10
268 dataset. Specifically, given the matrix factorization of the original image and reconstructed image:

269 $X_0 = \sum_{i=1}^{r} \sigma_{0,i} u_i v_i^\top$ and $\hat{X}_0 = \sum_{i=1}^{r} \hat{\sigma}_{0,i} \hat{u}_i \hat{v}_i^\top$ respectively, we compute $\frac{\left\| \sigma_{0,i} u_i v_i^\top - \hat{\sigma}_{0,i} \hat{u}_i \hat{v}_i^\top \right\|_2^2}{\left\| \sigma_{0,i} u_i v_i^\top \right\|_2^2}$

270 for $i = 1, 2, 3, 4, 5$ and models with 50%, 60%, 70%, 80%, 90% mask ratio, shown in Figure 2. Our
271 results demonstrate: 1. The reconstructed image is low-rank, since the distances are increasing with
272 index, which means that the leading components are more closely approximated. 2. A higher mask
273 ratio tends to have smaller number of clusters, as the slope is steeper when mask ratio is higher.

### 6.2 Visualizing cluster results of MAE features

275 In Section 4.3, we argued that MAE is a token level Contrastive Learning, which could result in
276 good zero-shot segmentation results. Here we demonstrate a few non-cherry-picked segmentation
277 examples by performing K-means on MAE features in Figure 2.

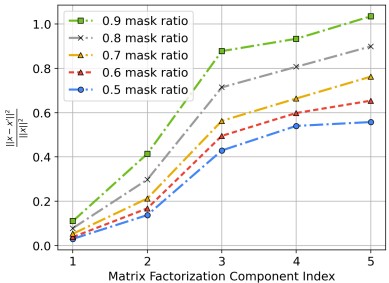

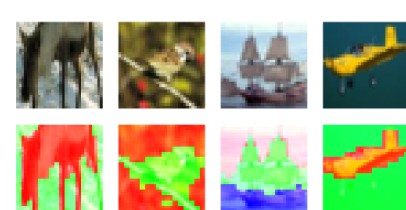

Figure 2: Distances between leading matrix factorization components of reconstructed image and original image.

Figure 3: K-means on MAE encoder features.

## 6.3 Different Probing Methods

In Table 1, we compare three different probing methods mentioned in Section 5.2. We conducted our experiments on both Cifar10 and IN-1K datasets with linear probing with a linear head (LP), non-learnable batch normalization + linear probing (BN + LP), and partial finetuning (Partial FT). For partial finetuning, we tune a linear head and the last qkv projection layer in the encoder, which is also linear.

We observe that a non-learnable BN layer can significantly improve the linear probing performance while a learnable token mixing layer can further improve the performance. The performance gain here is larger than what is typically seen in other Contrastive Learning models [9]. This phenomenon provides support for our assumption that the class token learned through the MIM pretext task may not accurately select clusters highly relevant to class labels. The incorporation of token mixture layers enables the reselection of clusters and leads to a performance boost.

| Dataset | LP | BN+LP | Partial FT |
|---------|------|------------|------------|
| Cifar10 | 64.4 | 76.6 (**+12.2**) | 83.3 (**+6.7**) |
| IN-1K | 48.0 | 68.0 (**+20.0**) | 69.3 (**+1.3**) |

Table 1: Classification accuracy with different probing methods.

| Heads | LP | FT |
|-------|------------|------------|
| 3 | 64.4 | 89.7 |
| 6 | 67.5 (**+3.1**) | 90.2 (**+0.5**) |

Table 2: Classification accuracy with different attention head numbers.

## 6.4 Ablation on Number of Attention Heads

This experiment supports our assertion in Section 5.3 that the number of attention heads can impact the performance of MIMs. We pretrained a ViT-Tiny on Cifar10 with 3 and 6 attention heads respectively and their downstream task accuracy with linear probing (without BN) and finetuning are shown in Table 2. By increasing the number of heads, we observed a significant improvement in accuracy for both linear probing and finetuning on the classification task.

# 7 Discussion

While mask modeling can be seen as a variant of Contrastive Learning at the token level, it is much different from traditional Contrastive methods. The primary distinction lies in the definition of positive and negative pairs: mask modeling methods derive these pairs from natural signals, whereas Contrastive methods use externally sourced human knowledge. This fundamental difference also impacts the data augmentation approach used in each method, with mask modeling employing weak augmentation, and Contrastive methods depending on strong augmentations. It thus raises the intriguing question of whether there could be a unified approach to the augmentation process. Furthermore, there's a compelling need to explore how mask models can be successfully employed in the arena of multimodal self-supervised learning, especially given the challenge that natural signals often do not align across different modalities.

# 8 Conclusion

In this paper, we propose a theoretical framework for analyzing mask models. We discover that mask modeling is a form of Contrastive Learning at the token level, which may account for its success. Our framework also addresses important questions regarding the behavior of mask models. We hope that our study will offer valuable insights into designing self-supervised learning algorithms and model architectures.

**Limitation** While our paper puts forth a theoretical framework to analyze mask models and elucidates their relationship with Contrastive Learning, we remain unable to provide a provable guarantee for the downstream performance of self-supervised pretraining models. The persisting challenge to fully comprehend Contrastive Learning - specifically, the inductive bias inherent in neural networks - continues to apply to mask models as well [30].

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
