# 1 Experiments Parameters

Table 1: **Pretraining parameters of ViT-T on Cifar10**

| config | value |
| --- | --- |
| optimizer | AdamW |
| base learning rate | 1.5e-4 |
| weight decay | 0.05 |
| optimizer momentum | $\beta_1, \beta_2 = 0.9, 0.95$ |
| batch size | 512 |
| learning rate schedule | cosine decay |
| warmup epochs [?] | 200 |
| total epochs | 2000 |
| augmentation | None |
| patch size | $2 \times 2$ |

Table 2: **Finetuning and Linear probing parameters of ViT-T on Cifar10**

| config | value |
| --- | --- |
| optimizer | AdamW |
| base learning rate | 1e-3 |
| weight decay | 0.05 |
| optimizer momentum | $\beta_1, \beta_2 = 0.9, 0.999$ |
| batch size | 128 |
| learning rate schedule | cosine decay |
| warmup epochs | 5 |
| training epochs | 100 |
| augmentation | None |

# 2 More MAE Segmentation examples

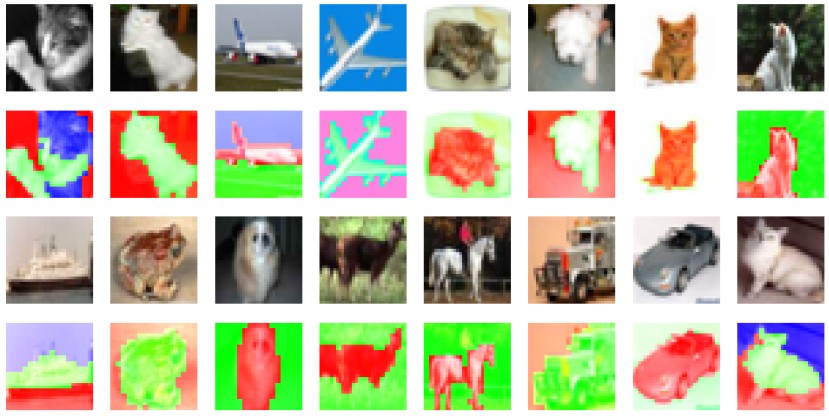