# OpenReview forum: "Mask Models are Token Level Contrastive Learners"
_ICLR.cc/2024/Conference — ICLR 2024 Conference Withdrawn Submission_

### Official Review · Reviewer_oRYb · 2023-10-28

**Soundness:** 3 good
**Presentation:** 3 good
**Contribution:** 3 good
**Rating:** 6
**Confidence:** 4

**Summary:**

This paper studies the underlying mechanism of the masked language model. Similar to the situations in contrastive learning, the authors propose to explain the success via the spectral clustering framework and establish the underlying clustering structure in the natural data is the key. The only difference is that pairs are from human heuristic in contrastive learning (i.e., pairs of data after augmentation), whereas here the pairs are from the natural data (i.e., a token and the surrounding data). The authors also provide some empirical evidence that validates their result.

**Strengths:**

- They extend the theory of spectral contrastive learning to the masked language model domain, which is a meaningful generalization of the original theory.
- The paper is well-written and the experimental details are stated clearly.

**Weaknesses:**

- The theory is largely the same as the original spectral contrastive learning paper despite different definitions of positive pairs. There seems to be a lack of theoretical innovation despite mainly being a theory paper.
- The experiments are largely on vision data. Would be great to include more experiments on language domain which is where masked token prediction is widely used.
- There's a lack of explanation of why the clustering structure appears in the first place.

**Questions:**

- Do you have any intuition on how much the inductive bias comes into play? Since the tokens of different locations are largely treated as different tasks in the spectral contrastive learning theory, it seems like there must be some inductive bias that makes the representations similar across different places.

---

### Official Review · Reviewer_rQ46 · 2023-10-29

**Soundness:** 3 good
**Presentation:** 2 fair
**Contribution:** 2 fair
**Rating:** 3
**Confidence:** 3

**Summary:**

This paper proposes a theoretical framework for understanding mask models and establishes the connections between mask models and spectral contrastive learning. Based on that, the paper explains some important designs in mask models. Empirically, the authors verify their assumptions and theoretical analysis (e.g., the choices of mask ratios, batch normalization, and backbone architecture are important in MAE models) on real-world datasets.

**Strengths:**

The theoretical analysis of mask models is important and under-explored. The connection between mask models and contrastive learning is interesting.

**Weaknesses:**

1. Based on the theoretical analysis in Section 4, the objective of mask models is equivalent to the spectral contrastive loss. However, I think there exists an obvious difference, i.e., the adjacent matrix in contrastive learning represents the probability that two samples can be selected as a positive pair while that in mask models represents the input similarity of two samples. And the discussions in this paper about that are not enough. For example, I do not think the models can learn meaningful representations if we select positive pairs in contrastive learning based on the input similarity.
2. I think there are too many detailed proofs in Section 4 and it is hard to capture the main messages derived in the Theorems.
3. I note that the theoretical analysis assumes the encode-decoder architecture is equivalent to the encoder in contrastive learning. However, in downstream tasks, we usually only use the encoder $f$. Is it possible to theoretically analyze the downstream performance?
4. The experiments in this paper verify the theoretical analysis. However, is it possible to provide new insights to further increase the performance of mask models based on the analysis?
5. $N$ denotes different meanings in the proofs of Theorem 4.3 and Lemma 4.6.

**Questions:**

see my comments above.

---

### Official Review · Reviewer_wnEL · 2023-10-31

**Soundness:** 1 poor
**Presentation:** 2 fair
**Contribution:** 2 fair
**Rating:** 3
**Confidence:** 3

**Summary:**

This paper provides a theoretical framework for understanding the working mechanism of masked image modeling (MIM). Specifically, this paper connects MIM with spectral clustering, through which MIM is viewed as a token-level contrastive learning. This view explains several intriguing empirical questions: 1) varying optimal masking ratios across domains, 2) the large performance gap between linear probing and fine-tuning, and 3) the interaction between MIM and model architectures.

**Strengths:**

The problem, understanding performant deep learning techniques from a theoretical perspective, provides insights into harnessing existing frameworks and developing novel approaches for practitioners. This work gives intuitions to certain behaviors of MIM as listed in the summary.

**Weaknesses:**

• The contributions over prior work are not fully discussed. For instance, the connection between spectral clustering and contrastive learning has already been studied in [1], whereas this paper is not discussed in the paper. In my opinion, there is a significant overlap between the messages (e.g., masking ratios), which largely weakens the current work's contribution.

• This paper lacks substance in theory. The major theory in this paper connects MIM and contrastive learning, which has already been given in [1]. Moreover, the remarks relating theory to MIM behaviors (i.e., Section 5) are somehow either too sloppy or straightforward. For instance, I don't see the logical connection between the number of clusters and the number of attention heads in line 256-257. I would appreciate either mathematical illustrations or more rigorous explanations for statements like this.

•  I find the experimental section highly underwhelming for a paper aiming to understand empirical behaviors. The results regarding MAE (Section 6.3 and 6.4) are already well known (finetuning > linear probing) and can be totally expected (more attention heads -> higher performance). They do not directly substantiate and are not specific to the theoretical insights.

[1] https://arxiv.org/abs/2210.08344

Some typos:

line 38: ".."

line 153: "are utilize"

**Questions:**

I would like to learn about the authors' response to the weaknesses listed above, which may give me a clearer perspective on the paper's contribution.

---

### Official Review · Reviewer_zZgx · 2023-11-01

**Soundness:** 2 fair
**Presentation:** 1 poor
**Contribution:** 2 fair
**Rating:** 3
**Confidence:** 2

**Summary:**

This paper conducts a theoretical study of masked image modeling (MIM) that relates the masked image modeling objective to a type of contrastive loss.  The theory shows that training a model to reconstruct the masked image patches is equivalent to minimizing a "token-wise" contrastive loss defined over the image patches.

In regular spectral contrastive loss, the adjacency matrix is based on similarity from external knowledge (e.g., certain data augmentations should result in similar representations). In this case, the adjacency matrix is based on the inner products between low-rank patch approximations in pixel space.

**Strengths:**

- Attempts to relate mask-reconstruction pretraining (typically thought of more as a generative pretraining method than a contrastive one) with contrastive losses, for which a number of theoretical results are known.

- The theory suggests intuitive ways of setting parameter choices for masked pretraining (Section 5).

**Weaknesses:**

- Possibly interesting results obscured by poor presentation. For example:

    - why is the input $X$ in the minimizations (1)/(5)? I thought the input was typically a random set of patches. Clearly the min over $X$ can typically be obtained by setting $X = X_0$. The MAE paper does not optimize over the input patches.
    - The token mixing layer definition is unclear and seems out of place. What does it mean to mix features on the patch level? Not defined here in a self-contained way.

    -  The construction in Section 4 is very unclear. What's the graph $G$? What are the nodes---patches? How are patches defined? How many possible patches are there? How is the rank $r$ defined?

    - How should I think about this adjacency matrix? Why is the inner product between low-rank patch approximations in pixel space the right way of thinking about patch similarity?

    - L158-159: _"Given the corresponding normalized adjacency matrix A, optimizing mask modeling is equivalent to optimize the following loss on classification downstream tasks"_

        Why "downstream tasks"? This looks like a pretraining loss.
    - There are numerous typos and grammatical errors throughout.

- The title seems like overselling. "Mask Models are Token Level Contrastive Learners" makes it sound like it also applies to BERT-style masking in NLP, but the motivation, assumptions, and details all heavily focus on the image modeling case. It should be clear that it's about masked *image* models

- Missing several relevant references, e.g. [1], [2]. The latter analysis is general and seems like it might apply to the masked modeling case as well.

- The predictions in Section 5 on the correct way of setting parameters like the masking ratio are not tested here empirically, and they rely on several untested assumptions (such as that $rank(g(f(X))$ is a constant proportional to the number of visible patches).


- The earlier theory does not explain the results in Section 5/6 on token mixing layers and batch normalization. There is no rigorous statement that connects back to Sections 3 and 4 showing that e.g. batch normalization layers will improve linear probing performance.

[1] Wei, Colin, Sang Michael Xie, and Tengyu Ma. "Why do pretrained language models help in downstream tasks? an analysis of head and prompt tuning." Advances in Neural Information Processing Systems 34 (2021): 16158-16170.

[2] Balestriero, Randall, and Yann LeCun. "Contrastive and non-contrastive self-supervised learning recover global and local spectral embedding methods." Advances in Neural Information Processing Systems 35 (2022): 26671-26685.

**Questions:**

(see above questions about presentation)